# Improving Photocatalytic Degradation Activity of Organic Pollutant by Sn^4+^ Doping of Anatase TiO_2_ Hierarchical Nanospheres with Dominant {001} Facets

**DOI:** 10.3390/nano9111603

**Published:** 2019-11-12

**Authors:** Meiling Sun, Weichong Kong, Yunlong Zhao, Xiaolin Liu, Jingyue Xuan, Yunyan Liu, Fuchao Jia, Guangchao Yin, Jun Wang, Junkai Zhang

**Affiliations:** 1School of Physics and Optoelectronic Engineering, Shandong University of Technology, Zibo 255000, China; sunml@sdut.edu.cn (M.S.); jiafuchao@sdut.edu.cn (F.J.); yingc@sdut.edu.cn (G.Y.); 2Key Laboratory of Functional Materials Physics and Chemistry of the Ministry of Education, Jilin Normal University, Siping 136000, China

**Keywords:** TiO_2_ hierarchical nanospheres, {001} facets, Sn^4+^ doping, solvothermal route, photocatalytic activity

## Abstract

Herein, high-energy {001} facets and Sn^4+^ doping have been demonstrated to be effective strategies to improve the surface characteristics, photon absorption, and charge transport of TiO_2_ hierarchical nanospheres, thereby improving their photocatalytic performance. The TiO_2_ hierarchical nanospheres under different reaction times were prepared by solvothermal method. The TiO_2_ hierarchical nanospheres (24 h) expose the largest area of {001} facets, which is conducive to increase the density of surface active sites to degrade the adsorbed methylene blue (MB), enhance light scattering ability to absorb more incident photons, and finally, improve photocatalytic activity. Furthermore, the Sn*_x_*Ti_1−*x*_O_2_ (STO) hierarchical nanospheres are fabricated by Sn^4+^ doping, in which the Sn^4+^ doping energy level and surface hydroxyl group are beneficial to broaden the light absorption range, promote the generation of charge carriers, and retard the recombination of electron–hole pairs, thereby increasing the probability of charge carriers participating in photocatalytic reactions. Compared with TiO_2_ hierarchical nanospheres (24 h), the STO hierarchical nanospheres with 5% *n*_Sn_/*n*_Ti_ molar ratio exhibit a 1.84-fold improvement in photodegradation of MB arising from the enhanced light absorption ability, increased number of photogenerated electron–hole pairs, and prolonged charge carrier lifetime. In addition, the detailed mechanisms are also discussed in the present paper.

## 1. Introduction

With the rapid development of industrialization and urbanization, the water pollution problem has gradually threatened human health and received worldwide attention, while how to effectively degrade the organic pollutants of waste water has been a key issue in recent years. Since the discovery of photocatalysis in 1976 [1], the semiconductor-based photocatalysts have played an important role in degrading pollutants and purifying water sources. Among a wide variety of photocatalysts, TiO_2_ as an eco-friendly semiconductor material is a good choice for the degradation of organic pollutants in water under the excitation of light due to its proper electronic band structure, excellent catalytic activity, good chemical stability, non-toxicity, low-cost, and photocorrosion resistance [2,3]. During the photocatalytic reaction, the photocatalysts absorb photons with energy higher than their band gaps and generate charge carriers, then the photoinduced electrons and holes diffuse to the surface of photocatalysts before they are recombined with each other and participate in the redox reaction to degrade the pollutants into harmless substances, such as H_2_O and CO_2_ [4]. However, it is reported that the band gap of TiO_2_ is very wide (3.0~3.2 eV) [5], which makes its light response region only in ultraviolet (UV) (<5% of total sunlight) [6]. Meanwhile, the low electron mobility (0.3 cm^2^V^−1^ s^−1^) [7] and fast photoinduced charge recombination rate (10^−12^ to 10^−11^ s) [8] reduce the utilization efficiency of charge carriers. Therefore, it is necessary to improve the light absorption ability, promote charge transmission, and suppress electron–hole recombination of TiO_2_ to boost the photocatalytic performance.

The optical absorption and charge transport properties of TiO_2_ crystals are determined not only by the physical and chemical properties of the material itself, but also by their morphology, structure, size, surface properties, and so on. In recent years, many TiO_2_ nanomaterials (e.g., nanoparticles, nanowires, nanobelts, nanosheets) [9,10,11] have been developed as photocatalysts. Among them, the TiO_2_ nanosheets with dominant {001} facets have attracted wide attention since Yang and colleagues first prepared them with 47% {001} facets [12]. The TiO_2_ nanosheet with dominant high-energy {001} facets has unique electronic band structure and surface structure characteristics. Firstly, it possesses a unique “surface heterojunction” that is formed by the {001} and {101} facets of TiO_2_ nanosheet owing to their different band edge positions, which can effectively promote the transport and separation of photogenerated charge carriers [13]. Secondly, the {001} facets possess higher surface energy (0.90 J m^−2^) [14], more unsaturated Ti atoms, and more active sites [15] than those of the general {101} and {010} facets, which will be more beneficial for the physical adsorption and chemical reaction of organic pollutants. Until now, plenty of investigations indicate that the TiO_2_ nanosheets with dominant {001} facets showed enhanced photodegradation efficiency compared to TiO_2_ nanoparticles [16,17]. However, most of the current TiO_2_ nanosheet photocatalysts are easily agglomerated powders, which will affect the exposed area of the high energy {001} facets, further limiting the photocatalytic performance. In addition, the TiO_2_ nanosheets with dominant high-energy {001} facets do not increase the light response range, and the light utilization is still very low. Therefore, further improving the dispersion, optimizing the exposure ratio of high-energy {001} facets, and enhancing the light harvesting ability are effective strategies to improve the photocatalytic performance of TiO_2_ materials.

Recently, doping with many metal ions (such as Fe, Nb, Ni, Co, etc.) [18,19,20,21,22] or non-metal ions (such as C, S, N, etc.) [23,24,25] to further improve the light and electrical properties of materials has attracted extensive attention. The proper chemical element doping on TiO_2_ could generate the donor or acceptor states, which will not only enhance its optical absorption ability, but also improve the separation and transmission of photogenerated charge carriers [26], thereby improving the photocatalytic activity of TiO_2_. However, so far, most of the doping studies have focused on the TiO_2_ nanomaterials without dominant {001} facets, because the chemical element doping may affect the growth process and structure of the exposed {001} facets in TiO_2_ nanomaterials. In the view of further improving the photocatalytic performance of TiO_2_, it is highly desirable to incorporate dopants into TiO_2_ nanomaterials with dominant high-energy {001} facets. Among various dopants, Sn^4+^ will be a good choice for doping on TiO_2_ due to its proper ion radius compared to that of Ti^4+^ and the similar lattice structure between SnO_2_ and TiO_2_ [27,28], which will lead to good structural compatibility and stability of Sn^4+^ doped TiO_2_ (STO) nanomaterials. However, how does Sn^4+^ doping affect the growth and structure of the exposed {001} facets in TiO_2_ nanomaterials, what synergistic effect will Sn^4+^ doping and high-energy {001} facets have on the photocatalytic performance of TiO_2_ nanomaterials, and which doping ratio of Sn^4+^ is optimal to enhance the photocatalytic activity of TiO_2_ nanomaterials with dominant {001} facets, and so on, so many interesting subjects need to be investigated systematically due to the lack of the relevant research in the past.

Herein, we first studied the synthesis of Sn^4+^ doped and nanosheet-based TiO_2_ hierarchical nanospheres with dominant high-energy {001} facets by a facile one-pot solvothermal method for greatly improving their surface characteristics, photon absorption, and charge transport, then enhancing their photocatalytic activity. By tuning reaction time, the growing process, structural evolution mechanism, and photocatalytic activity of TiO_2_ hierarchical nanospheres were all systemically investigated. Results indicated that when the reaction time was 24 h, the TiO_2_ hierarchical nanospheres owned the largest area of {001} facets and showed the optimal photodegradation activity of methylene blue (MB). Furthermore, after Sn^4+^ doping, the mixed-cation hierarchical nanospheres of Sn*_x_*Ti_1−*x*_O_2_ (STO) were fabricated, in which the Sn^4+^ doping energy level and surface hydroxyl group had effectively broadened the light absorption range, promoted the generation of electron–hole pairs, and retarded the recombination of excited electrons and holes, thereby increasing the probability of photogenerated charge carriers participating in photocatalytic reactions. Therefore, the STO hierarchical nanospheres exhibited significantly enhanced photodegradation activity compared to the pristine TiO_2_ hierarchical nanospheres.

## 2. Materials and Methods 

### 2.1. Materials

Titanium (IV) isopropoxide, diethylenetriamine, isopropyl alcohol (IPA), ethanol, SnCl_4_·5H_2_O and MB were purchased from Sinopharm Chemical Reagent Co., Ltd. (Shanghai, China). Benzoquinone (BQ) was purchased from Maclin Reagent Co., Ltd. (Shanghai, China). All reagents employed were of analytic grade and without further purification.

### 2.2. Preparation of TiO_2_ Hierarchical Nanospheres 

In this experiment, the TiO_2_ hierarchical nanospheres with dominant high-energy {001} facets were prepared by a facile one-step solvothermal method. The schematic illustration of experimental procedure is shown in Figure 1. First, 0.05 mL diethylenetriamine was added to 71 mL of IPA. After stirring for a few minutes, 3.35 mL (10 mmol) of titanium (IV) isopropoxide was added to the mixed solution, and then the resulting solution was ultrasonicated for another 5 min. Subsequently, the prepared solution was transferred into a sealed Teflon-lined autoclave (100 mL), and the solvothermal synthesis was carried out at 200 °C under different reaction times (6 h, 12 h, 24 h, and 48 h). Finally, the samples were harvested via centrifugation, washed thoroughly with ethanol, and dried at 60 °C for 24 h. Then all samples were annealed at 400 °C for 2 h with a heating rate of 1 °C min^−1^ to increase the crystallinity.

### 2.3. Preparation of STO Hierarchical Nanospheres 

The synthetic method of STO hierarchical nanospheres with dominant high-energy {001} facets is similar to that of TiO_2_ hierarchical nanospheres (see Figure 1). First, 0.05 mL of diethylenetriamine was added to 71 mL of IPA and the mixture was stirred for a few minutes. Then 3.35 mL of titanium (IV) isopropoxide and different amounts (0.1 mmol, 0.5 mmol, 1 mmol, and 1.5 mmol) of SnCl_4_·5H_2_O were added to the mixed solution. The mixture was ultrasonicated for another 5 min and then transferred into a sealed Teflon-lined autoclave (100 mL). The solvothermal synthesis was carried out at 200 °C for 24 h. The following treatment method of the samples is the same as that used to synthesize TiO_2_ hierarchical nanospheres.

### 2.4. Characterization

The crystalline phase of the samples was checked by X-ray power diffraction (XRD, Rigaku D/max-2500, Tokyo, Japan) with CuKα radiation (λ = 1.5418 Å). Chemical composition and elemental chemical status of samples were examined by X-ray photoelectron spectroscopy (XPS, ESCALAB 250, West Sussex, UK) with a focused Al Kα radiation (1486.6 eV). Morphology and structures of samples were investigated by field emission scanning electron microscopy (FESEM, JEOL JSM-6700F, Tokyo, Japan), transmission electron microscopy (TEM), and high-resolution transmission electron microscopy (HRTEM, JEOL JEM-2200FS, Tokyo, Japan). The UV-vis absorption spectra of samples were measured by a double-beam spectrophotometer (Shimadzu UV-3600, Kyoto, Japan). Photoluminescence (PL) spectra were conducted by a fluorescence spectrophotometer (Tianjin Gangdong F-380, Tianjin, China) with an excitation wavelength of 320 nm. Total organic carbon (TOC) value was obtained with the total organic carbon analyzer (Shimadzu TOC-L, Kyoto, Japan).

### 2.5. Photocatalytic Activity Tests

The photocatalytic activity of TiO_2_ and STO hierarchical nanospheres were examined by the degradation of MB. A 500 W Xe lamp with an illumination wavelength from 280 nm to 980 nm was used as the light source and placed 15 cm above the solution surface. First, 25 mg of TiO_2_ or STO photocatalysts were added into a quartz reactor containing 100 mL MB aqueous solution (5 mg/L) and magnetically stirred in darkness for 30 min to reach adsorption–desorption equilibrium before irradiation. Upon irradiation, 5 mL of the suspension was collected and centrifuged per 20 min. The degradation activity of MB was analyzed by a UV-3600 spectrophotometer at its absorption maximum (664 nm). In addition, more tests were done to exclude the sensitization of MB, to study the influence of the medium, and to check the loading effect of catalysts during the photodegradation processes.

### 2.6. Radical Trapping Experiments

The radical trapping experiments were conducted to investigate the active radical species during the photodegradation processes of MB. It is well-known that the main reactive species are the superoxide radical (O_2_^−^) and hydroxyl radicals (·OH) in the photodegradation processes, so the IPA and BQ were chosen as the radical scavengers for ·OH and O_2_^−^, respectively. The experimental procedure was similar to the photocatalytic activity tests (Section 2.5), except with the addition of the radical scavengers (1.0 mmol/L) into the photocatalytic solution.

## 3. Results and Discussion

### 3.1. Characterizations of TiO_2_ Hierarchical Nanospheres

The FESEM images of the as-prepared TiO_2_ hierarchical nanospheres under different solvothermal reaction times are shown in Figure 2. It can be seen clearly that all samples are composed of many TiO_2_ nanosheets and have uniform size, good dispersibility, and three-dimensional (3D) hierarchical structure. In detail, when the reaction time is 6 h (Figure 2a), many TiO_2_ nanosheets are formed and self-assemble into the 3D hierarchical nanospheres with an average diameter of ~200 nm, but the TiO_2_ nanosheets are assembled very loosely and there is a lot of space between them. Increasing the reaction time to 12 h (Figure 2b), the diameter of the TiO_2_ hierarchical nanospheres increases to ~550 nm. The TiO_2_ nanosheets are assembled more tightly, but the space between the TiO_2_ nanosheets is filled by many nanoparticles. When the reaction time reaches 24 h (Figure 2c), all the TiO_2_ nanoparticles filled in the space between the nanosheets and regrow into the new smooth nanosheets, and the diameter of the TiO_2_ hierarchical nanospheres increases into ~800 nm. Simultaneously, there is adequate space to expose more high-energy {001} facets between the TiO_2_ nanosheets. However, when the reaction time exceeds 24 h to 48 h (Figure 2d), the diameter of TiO_2_ hierarchical nanospheres no longer increases, but the surface of the TiO_2_ nanosheets becomes rough. It is likely that the solution is completely reacted after 24 h, so no new TiO_2_ nanosheets can be formed and self-assemble into TiO_2_ hierarchical nanospheres. Meanwhile, as the TiO_2_ nanospheres have been immersed in the solution for a long time, the etching effect will make part of TiO_2_ nanosheets dissolve. From the point view of morphology, the 24 h obtained TiO_2_ hierarchical nanospheres will have better photocatalytic performance.

Through the above experimental phenomena, the formation mechanism of 3D hierarchical nanospheres is shown in Figure 3. During the solvothermal reaction, the TiO_2_ nanosheets are first formed, as shown in Figure 3a. As the number of TiO_2_ nanosheets increases, they self-assembled to form 3D hierarchical nanospheres (Figure 3b) due to the Ostwald ripening upon the solvothermal treatment. With the increasing reaction time, the nanosheets continue to grow and form TiO_2_ hierarchical nanospheres with a larger diameter. Meanwhile, many new nanoparticles are formed, reducing the space between the TiO_2_ nanosheets, as shown in Figure 3c. As the reaction time continues to increase, the nanoparticles regrow into new nanosheets, and the diameter of the TiO_2_ hierarchical nanospheres further increases. When the solution is completely reacted, the diameter of hierarchical nanospheres no longer increases. At this point, the TiO_2_ hierarchical nanospheres are more evenly distributed and own enough space to expose more surface area for adsorbing organic pollutants during the catalytic process, as shown in Figure 3d. Finally, when the reaction time is further increased (Figure 3e), the dissolution reaction takes the leading position and the TiO_2_ hierarchical nanospheres are etched, which will lower the photocatalytic properties of samples.

The crystal structure, phase composition, and microstructure of TiO_2_ hierarchical nanospheres (24 h) were determined by XRD, TEM, and HRTEM measurements. It is clear from the XRD pattern (Figure 4a) that the main diffraction peaks of TiO_2_ hierarchical nanospheres at 25.29°, 37.82°, 48.05°, 53.91°, 55.07°, 62.70°, 68.77°, 70.32°, and 75.03° are indexed to the (101), (004), (200), (105), (211), (204), (116), (220), and (215) planes of anatase phase TiO_2_, respectively (JCPDS No. 21-1272) Furthermore, the diffraction peaks of other crystalline products have not been found, indicating the high purity and crystallinity of as-prepared TiO_2_ hierarchical nanospheres. The typical TEM image of samples indicates that the TiO_2_ nanosphere has a hierarchical structure (Figure 4b). From the light-colored area at the edge of the TiO_2_ hierarchical nanosphere, it is clear that the nanosphere is self-assembled from the visible nanosheets, consistent with the FESEM images. Figure 4c shows the corresponding HRTEM image of a single TiO_2_ nanosheet. The observed spacing fringe of 0.190 nm is attributed to the mutually perpendicular (020) and (200) lattice plane of anatase TiO_2_, which demonstrates that the top and bottom surfaces of the TiO_2_ nanosheets are high-energy {001} facets [29]. That is to say, the obtained TiO_2_ hierarchical nanospheres mainly exposed the high-energy {001} facets.

The photocatalytic properties of TiO_2_ hierarchical nanospheres prepared under different solvothermal reaction times were investigated through the photodegradation of MB under Xe lamp irradiation, as shown in Figure 5, where C is the concentration of MB at irradiation time t and C_0_ is the initial concentration of MB. In the dark adsorption process, the TiO_2_ hierarchical nanospheres prepared under different times almost exhibit the same adsorption capacity for MB. Under light irradiation, it is clear that the TiO_2_ hierarchical nanospheres prepared at 6 h degrade 42.1% MB within 100 min. However, as the solvothermal time increases, the photocatalytic degradation activity of TiO_2_ hierarchical nanospheres first increases and then decreases. When the solvothermal time is 24 h, the TiO_2_ hierarchical nanospheres degraded up to 50.6% MB. The initial enhancement of photocatalytic degradation activity might because the diameter of the TiO_2_ hierarchical nanospheres gradually increases and more high-energy {001} facets are exposed with the increase of solvothermal reaction time (Figure 2), which increases the number of surface active sites and improves light scattering ability, thereby enhancing the surface adsorption ability of MB, absorbing more incident photons to generate more electron–hole pairs and finally enhancing the degradation of adsorbed MB. However, when the solvothermal reaction time exceeds 24 h, the TiO_2_ hierarchical nanospheres will be etched, resulting in the decrease in photocatalytic degradation activity.

### 3.2. Characterizations of STO Hierarchical Nanospheres

To further improve the photocatalytic degradation properties of TiO_2_ hierarchical nanospheres with the optimal solvothermal time (24 h), we treated the hierarchical nanospheres with Sn^4+^ doping, and studied the effects of Sn^4+^ doping on the photocatalytic degradation activity of TiO_2_ hierarchical nanospheres. The FESEM images of STO hierarchical nanospheres with different Sn^4+^ doping concentration are shown in Figure 6. When the molar ratio of *n*_Sn_/*n*_Ti_ is 1% (Figure 6a), the STO nanospheres are still a 3D hierarchical structure composed of nanosheets, and the surface of the nanosheets is very smooth. Furthermore, the diameter of the STO hierarchical nanospheres is the same as that of TiO_2_ hierarchical nanospheres (24 h) (Figure 2c). As the molar ratio of *n*_Sn_/*n*_Ti_ increases to 5% (Figure 6b), the STO hierarchical nanospheres retain a similar 3D hierarchical structure, but the surface of the nanosheets becomes rough due to the growth of many nanoparticles and the diameter of the hierarchical nanospheres is increased to ~900 nm. Further increasing the molar ratio of *n*_Sn_/*n*_Ti_ to 10% (Figure 6c), the thickness of the nanosheets constituting the STO hierarchical nanospheres is increased and the space between the STO nanosheets is filled by many nanoparticles, which will reduce the specific surface area of the STO hierarchical nanospheres. Finally, when the molar ratio of *n*_Sn_/*n*_Ti_ is further increased to 15% (Figure 6d), the surface of the STO hierarchical nanospheres is filled by many nanoparticles and only a few STO nanosheets are exposed. It can be seen that as the Sn^4+^ doping concentration increases, the proportion of nanosheets in the STO hierarchical nanospheres will be gradually decreased, which will reduce the specific surface area of the samples. 

The XRD measurement of TiO_2_ (24 h) and STO hierarchical nanospheres were carried out to further explore the possible phase changes of TiO_2_ hierarchical nanospheres after Sn^4+^ doping with different concentrations, and the results are shown in Figure 7a. It is obvious that the spectra of all STO hierarchical nanospheres (curves b–e) are not significantly different from that of the TiO_2_ hierarchical nanospheres, and all of the diffraction peaks can be well indexed to the standard anatase structure of TiO_2_ (JCPDS No. 21-1272). This proved that when the molar ratio of *n*_Sn_/*n*_Ti_ is not more than 15%, Sn^4+^ doping will not change the crystal structure of TiO_2_. In more detail, we cannot see any other characteristic peaks of impurity phases such as Sn or SnO_2_ from the XRD patterns, indicating the high purity of the as-prepared STO hierarchical nanospheres. However, as can be seen from the magnified patterns in the 2*θ* range of 20°–30° (Figure 7b), the diffraction peaks of the STO hierarchical nanospheres (curves b–e) have an obvious small-angle shift compared with that of the original TiO_2_ hierarchical nanospheres (curve a), and the shifting angle of the diffraction peak gradually increases with the increasing molar ratio of *n*_Sn_/*n*_Ti_. This phenomenon should be attributed to the fact that the ionic radius of Sn^4+^ (0.69 Å) is similar to that of Ti^4+^ (0.53 Å), therefore Sn^4+^ easily occupies the site of Ti^4+^ in the TiO_2_ lattice, thereby forming the substituted doping structure (Sn*_x_*Ti_1−*x*_O_2_). When the Sn^4+^ occupies the site of Ti^4+^, it causes a slight lattice expansion of the STO samples because the ionic radius of Sn^4+^ is slightly larger than that of Ti^4+^ (Figure 7c). Therefore, the XRD diffraction peaks of the STO hierarchical nanospheres will shift to the smaller 2*θ* values.

The detailed microstructure of the STO hierarchical nanospheres was further examined by TEM and HRTEM, as shown in Figure 8a,b. It can be seen that the STO hierarchical nanospheres are composed of many nanosheets and possess high crystallinity. Moreover, the observed 0.191 nm spacing fringe in Figure 8b is consistent with the (200) plane of anatase TiO_2_, which is slightly larger than the lattice spacing of the original TiO_2_ hierarchical nanospheres (Figure 4c) because of the lattice expansion caused by the substitution of some Ti^4+^ sites by Sn^4+^. This further proves that Sn^4+^ doping has no effect on the crystal structure of TiO_2_ nanosheets, and the exposed crystal planes of STO hierarchical nanospheres are still dominated by high-energy {001} facets. Figure 8c shows the corresponding energy-dispersive X-ray (EDX) elemental mapping of an STO hierarchical nanosphere, which indicates that the Ti, O, and Sn elements are distributed uniformly throughout the whole STO hierarchical nanosphere. All these results confirm that the Sn^4+^ has been uniformly doped into the lattice of the TiO_2_ hierarchical nanospheres.

To further explore the effect of Sn^4+^ doping on the surface lattice of STO hierarchical nanospheres, the XPS measurement was carried out to investigate the change of surface composition and chemical status of as-prepared samples. Figure 9 shows that the high resolution XPS spectra of TiO_2_ (24 h) and STO hierarchical nanospheres consists of Sn 3d, Ti 2p, and O 1s. For the pristine TiO_2_ hierarchical nanospheres, the Sn signal has not been detected (Figure 9a). However, Figure 9b shows the Sn 3d XPS spectra, in which two obvious peaks at 486.5 eV and 495.1 eV were attributed to Sn 3d_5/2_ and Sn 3d_3/2_, respectively, corresponding to the Sn^4+^ state. The results confirm that Sn^4+^ replaces Ti^4+^ in the TiO_2_ bulk, forming a Ti–O–Sn substituted doping structure. The Ti 2p spectra of pristine TiO_2_ and STO hierarchical nanospheres are shown in Figure 9c,d, respectively. It can be seen that the Ti 2p_3/2_ and Ti 2p_1/2_ peaks of pristine TiO_2_ are centered at 458.2 and 463.9 eV, but those of the STO samples shift to higher binding energy (458.5 and 464.1 eV). This phenomenon might be attributed to the fact that the electronegativity of Sn ions in the Ti–O–Sn structure is greater than that of Ti ions in the Ti–O–Ti structure [30]. Therefore, the Sn^4+^ doping on the Ti^4+^ site will lead to a high-value shift of binding energy [30]. In addition, the shift of binding energy can also be observed in the O 1s spectra of the TiO_2_ and STO samples, as shown in Figure 9e,f. The O 1s peaks centered at about 529.4 eV, 531.2 eV (Figure 8e) and 529.7 eV, 532.3 eV (Figure 8f), can be assigned to the lattice oxygen (Ti–O bonds) and the surface hydroxyl groups (O–H bonds), respectively [31,32]. It is clear that the O 1s binding energies of STO are slightly higher than those of pristine TiO_2_. This is because the Sn^4+^ occupies the site of Ti^4+^ in the TiO_2_ lattice, meanwhile the electronegativity of Sn ions is greater than that of Ti ions [30]. Moreover, the surface O–H bonds’ total oxygen content (31%) in STO is higher compared to that in pure TiO_2_ (23%), indicating that the surface hydroxyl group content of STO is greater than that of pure TiO_2_. 

The UV-vis absorption measurements were performed on the TiO_2_ and STO hierarchical nanospheres as a function of Sn^4+^ doping concentration to evaluate the effect of Sn^4+^ doping on the light absorption properties of the samples. It can be seen from Figure 10 that the light absorption intensities of the STO hierarchical nanospheres (curves b–e) are higher than that of the pristine TiO_2_ hierarchical nanospheres (curve a), and the absorption edges of the STO samples (curves b–e) shift toward longer wavelengths (370 nm) compared to the pristine TiO_2_ UV-vis pattern (355 nm). These phenomena should be due to the fact that Sn^4+^ doping forms an impurity level at 0.38 eV below the conduction band (CB) of TiO_2_ [33], which can promote the excitation of electrons in a subsection from the valence band (VB) to the impurity level and then to the CB. The STO hierarchical nanospheres need to absorb light energy during each electronic transition, resulting in the increase in their light absorption intensities. Meanwhile, the energy level width of electron transition in STO hierarchical nanospheres is smaller than the band gap of TiO_2_, therefore Sn^4+^ doping broadens the light absorption range of samples. However, among the STO hierarchical nanospheres, with the increase of Sn^4+^ doping concentrations, the wavelength of absorption edges first increase and then decrease. This phenomenon should be ascribed to the fact that the Sn^4+^ doping will significantly improve the light absorption capacity of STO hierarchical nanospheres, however, the proportion of nanosheets in the hierarchical nanospheres gradually decreases as the Sn^4+^ doping concentration increases (Figure 6), which will reduce the light scattering ability of the samples. Therefore, when the molar ratio of *n*_Sn_/*n*_Ti_ is 5%, the STO hierarchical nanospheres exhibit the greatest absorption edge with a value of 370 nm. In addition, the PL spectra of TiO_2_ (24 h) and STO (0.5 mmol) hierarchical nanospheres was measured to further explore the effect of Sn^4+^ doping on the separation of electron–hole pairs, as shown in Appendix A. It is clear that the PL intensity of STO (0.5 mmol) is obviously lower than that of TiO_2_ (24 h), indicating that STO (0.5 mmol) has a better separation efficiency of electron–hole pairs than TiO_2_ (24 h), which is very beneficial for improving the photocatalytic performance [34,35].

The effective photoabsorption plays the salient role in enhancing the photon-to-charge conversion [36,37], which prompts us to study the photocatalytic properties of pristine TiO_2_ (24 h) and STO hierarchical nanospheres with different Sn^4+^ doping concentrations, as shown in Figure 11a. In the dark adsorption process, the adsorption capacities of TiO_2_ and STO hierarchical nanospheres for MB are basically the same. However, under light irradiation the STO hierarchical nanospheres display significantly enhanced photocatalytic activity for MB than pure TiO_2_ hierarchical nanospheres, indicating that Sn^4+^ doping plays an important role in improving the degradation activity of photocatalysts. Furthermore, the photocatalytic activity of STO hierarchical nanospheres for MB increase first and then decrease with the increase of Sn^4+^ doping concentration, and the STO hierarchical nanospheres with a 5% molar ratio of *n*_Sn_/*n*_Ti_ (0.5 mmol) exhibit the highest degradation of MB with a value of 93% after irradiation for 100 min, which is about 1.84-fold higher than that of pure TiO_2_ (24 h) hierarchical nanospheres. These results might be ascribed to the fact that the mixed-cation composition of Sn_x_Ti_1-x_O_2_ is formed in the lattice of STO, which will increase the light absorption range and facilitate the generation, separation, and transmission of the charge carriers [38], thus increasing the number of photogenerated carriers and improving the probability of photogenerated carriers participating in the photocatalytic reaction. However, when the molar ratios of *n*_Sn_/*n*_Ti_ are increased to more than 5%, the proportion of nanosheets in the hierarchical nanospheres gradually decrease as Sn^4+^ doping concentration increases, which will reduce the exposed area of high-energy {001} facets in the STO photocatalysts, thereby decreasing the number of active sites for the adsorption of MB and the dissociation of adsorbed MB. Meanwhile, the absorption edges of STO hierarchical nanospheres with more than a 5% *n*_Sn_/*n*_Ti_ molar ratio will be blue-shifted compared to that of STO samples with a 5% *n*_Sn_/*n*_Ti_ molar ratio, which will reduce the light absorption properties of the photocatalyst. Lastly, excess Sn^4+^ in STO hierarchical nanospheres can act as recombination centers, which will be detrimental to the photodegradation activity of STO photocatalysts. Therefore, the STO (0.5 mmol) hierarchical nanospheres exhibit the best photodegradation efficiency for MB. In addition, the TOC value of the solution after the photodegradation by STO (0.5 mmol) hierarchical nanospheres for 100 min can decrease to 26% of the original value, which implies the effective decomposition of the organic species [39].

To further explore the photocatalytic reaction mechanism and identify the primary active radical species during the photodegradation processes, the radical trapping experiments were conducted using IPA as the scavenger for ·OH and BQ as the scavenger for O_2_^−^, as shown in Figure 11b. It can be seen clearly that the photodegradation of MB remarkably decreases after that IPA is added as a ·OH scavenger, which indicates that ·OH is the primary active radical species during the photocatalytic degradation processes of MB. Compared with IPA, the photodegradation of MB also diminished after the addition of BQ as an O_2_^−^ scavenger, but not as so much as that of IPA, which suggests that the O_2_^−^ radicals play a more minor role than the ·OH radicals during the photodegradation processes of MB [34,35]. In addition, the photocatalytic tests using the best STO (0.5 mmol) hierarchical nanospheres with various amounts (15, 25, and 35 mg) were done to investigate the loading effect of catalysts for the degradation of MB (see Appendix A). The results indicate that the STO (0.5 mmol) hierarchical nanospheres show the best photocatalytic performance at the loading of 25 mg. The overloading of catalysts will negatively affect the degradation efficiency of MB, possibly because of the agglomeration of catalysts leading to a reduction in active sites on the surface of catalysts [35]. 

The possible mechanism of enhanced photocatalytic activity of STO hierarchical nanospheres with a 5% molar ratio of *n*_Sn_/*n*_Ti_ has also been investigated, as shown in Figure 12. Under the illumination of light, the electrons are transported from the VB to CB of TiO_2_, leaving holes in the VB. The photogenerated electrons will react with O_2_ to form O_2_^−^, while the photogenerated holes will react with H_2_O and OH^-^ to form the ·OH, then the O_2_^−^ and ·OH will further degrade or oxidize the MB on the photocatalyst surface [4]. It can be seen that increasing the specific surface area, enhancing the generation efficiency of charge carriers, and reducing the recombination rate of photogenerated electron–hole pairs will be beneficial to improve photodegradation activity of TiO_2_ photocatalysts. Therefore, the improvement of the photocatalytic activity of STO hierarchical nanospheres with a 5% molar ratio of *n*_Sn_/*n*_Ti_ is mostly attributed to the following four aspects. (1) The exposed highly reactive {001} facets of STO hierarchical nanospheres with an appropriate molar ratio of *n*_Sn_/*n*_Ti_ (5%) is similar to that of original TiO_2_ samples, which will provide more active sites to degrade the adsorbed MB; (2) Compared with the original TiO_2_ hierarchical nanospheres, the absorption spectrum of the STO samples has been broadened due to the Sn^4+^ doping, which greatly raised the utilization rate of solar energy, thereby generating more electron–hole pairs; (3) Since the Sn^4+^ doping level is 0.38 eV under the CB [36], the photogenerated electrons can be directly transferred to the surface of the photocatalyst through the Sn^4+^ doping level, resulting in efficient separation of photogenerated charge carriers; (4) After Sn^4+^ doping, the surface hydroxyl group content of STO is greater than that of pure TiO_2_, which will effectively prevent the recombination of photogenerated electron–hole pairs, prolong the carrier lifetime, improve the separation efficiency of photogenerated charge carriers and increase the probability of photogenerated charge carriers participating in photocatalytic reactions [40], thereby leading to an excellent photocatalytic performance.

## 4. Conclusions

In summary, we have studied the synthesis of Sn^4+^ doped and nanosheet-based TiO_2_ hierarchical nanospheres with dominant high-energy {001} facets by a facile one-pot solvothermal method for greatly improving their surface characteristics, photon absorption, and charge transport, thereby enhancing their photocatalytic activity. By tuning reaction time, the TiO_2_ hierarchical nanospheres (24 h) exposed the largest area of high-energy {001} facets, which is conducive to increasing the number of surface active sites to adsorb and degrade MB, enhance the light scattering ability to absorb more incident photons, and finally improve the photocatalytic degradation of MB to 50.6% within 100 min. Remarkably, after Sn^4+^ doping, the Sn^4+^ doping energy level and surface hydroxyl group of STO hierarchical nanospheres effectively broaden the light absorption range, generate more electron–hole pairs, and prevent the recombination of photogenerated charge carriers, thereby increasing the probability of photogenerated charge carriers participating in photocatalytic reactions. The STO hierarchical nanospheres with a 5% *n*_Sn_/*n*_Ti_ molar ratio degrade 93% MB after irradiation for 100 min, which is about 1.84 times higher than that of pristine TiO_2_ hierarchical nanospheres due to the enhanced light absorption ability, increased number of photogenerated electron–hole pairs, and extended photo-generated carrier lifetime. This work highlights the synergistic effects of the high-energy {001} facets and Sn^4+^ doping on the photocatalytic properties of TiO_2_ hierarchical nanospheres, which may be well suitable for improving other TiO_2_ materials in a variety of solar energy driven applications such as water-splitting, solar cells, and photodetectors.

## Figures and Tables

**Figure 1 nanomaterials-09-01603-f001:**
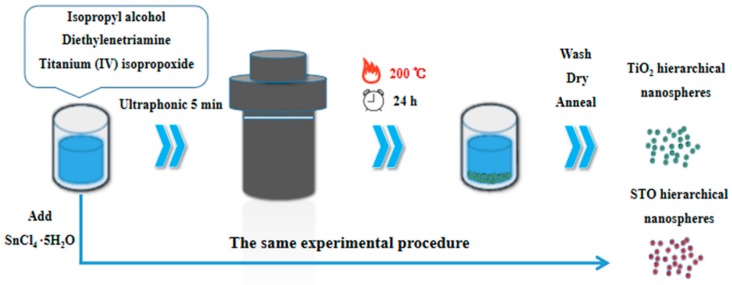
Schematic illustration of experimental procedure.

**Figure 2 nanomaterials-09-01603-f002:**
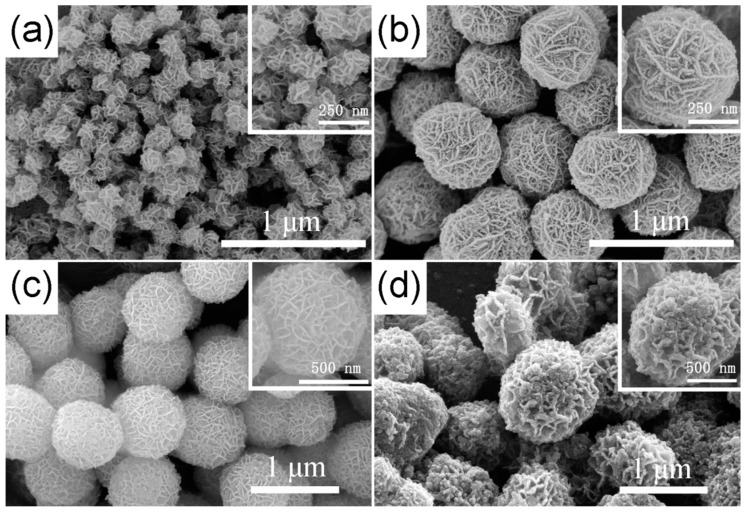
The FESEM images of the as-prepared TiO_2_ hierarchical nanospheres under different solvothermal reaction times: (**a**) 6 h, (**b**) 12 h, (**c**) 24 h, and (**d**) 48 h.

**Figure 3 nanomaterials-09-01603-f003:**
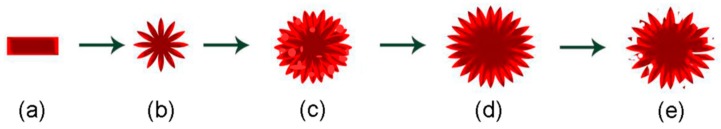
Illustration depicting the formation mechanism of 3D hierarchical nanospheres, (**a**–**e**) represents the evolution of material morphology with the growth time.

**Figure 4 nanomaterials-09-01603-f004:**
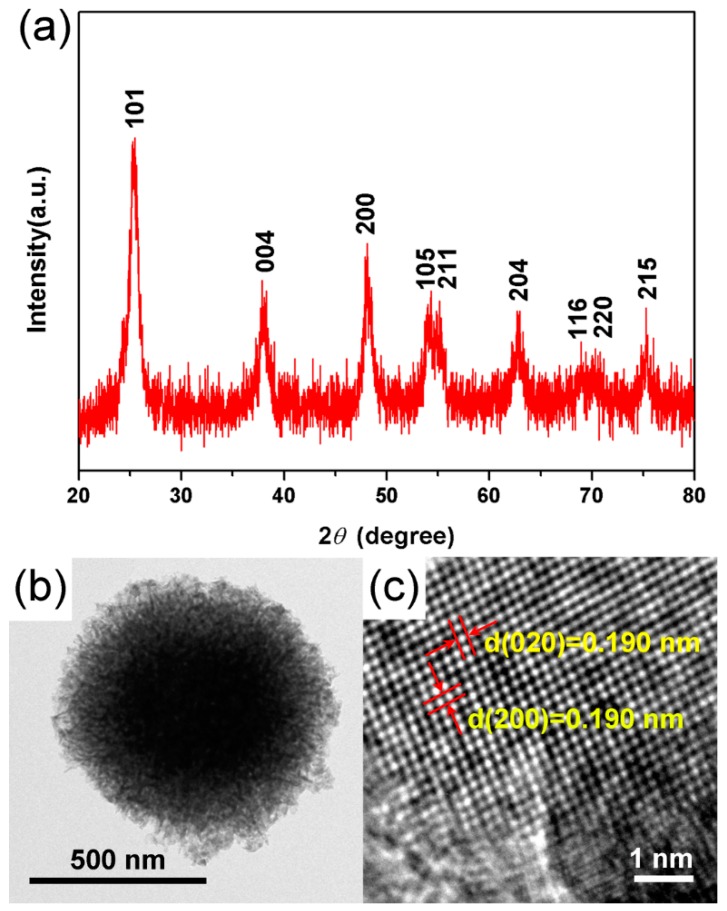
(**a**) XRD, (**b**) TEM, and (**c**) HRTEM images of the TiO_2_ hierarchical nanospheres prepared for 24 h.

**Figure 5 nanomaterials-09-01603-f005:**
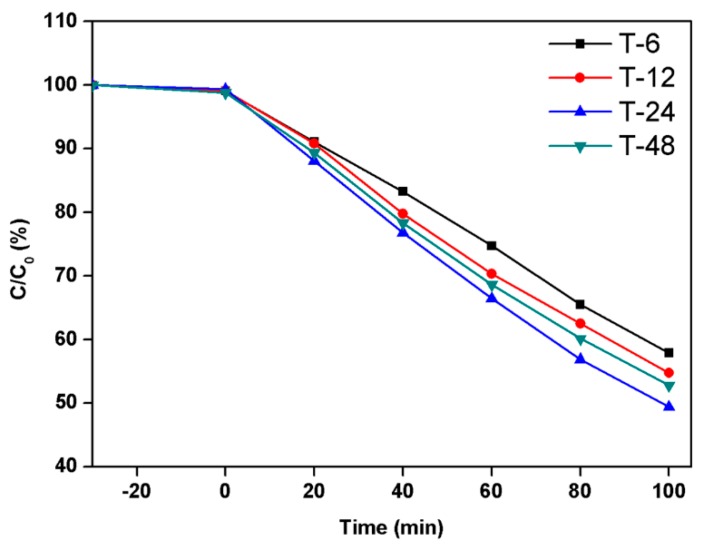
The photocatalytic activity of the TiO_2_ hierarchical nanospheres prepared under different solvothermal reaction times: 6 h, 12 h, 24 h, and 48 h.

**Figure 6 nanomaterials-09-01603-f006:**
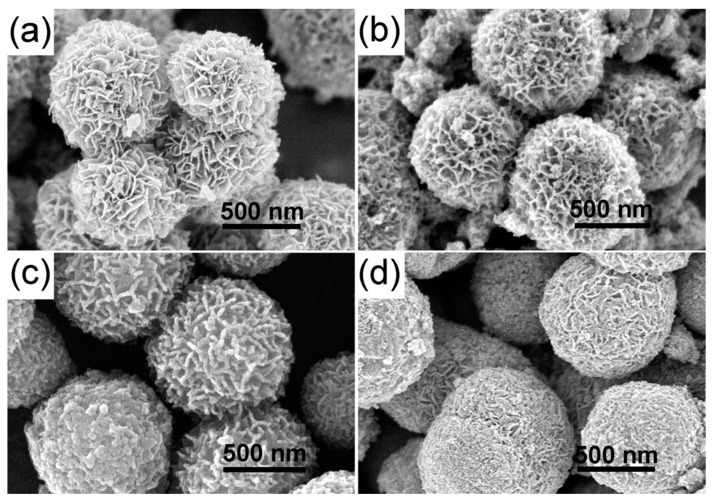
The FESEM images of the STO hierarchical nanospheres with different molar ratios of *n*_Sn_/*n*_Ti_: (**a**) 1%, (**b**) 5%, (**c**) 10%, and (**d**) 15%.

**Figure 7 nanomaterials-09-01603-f007:**
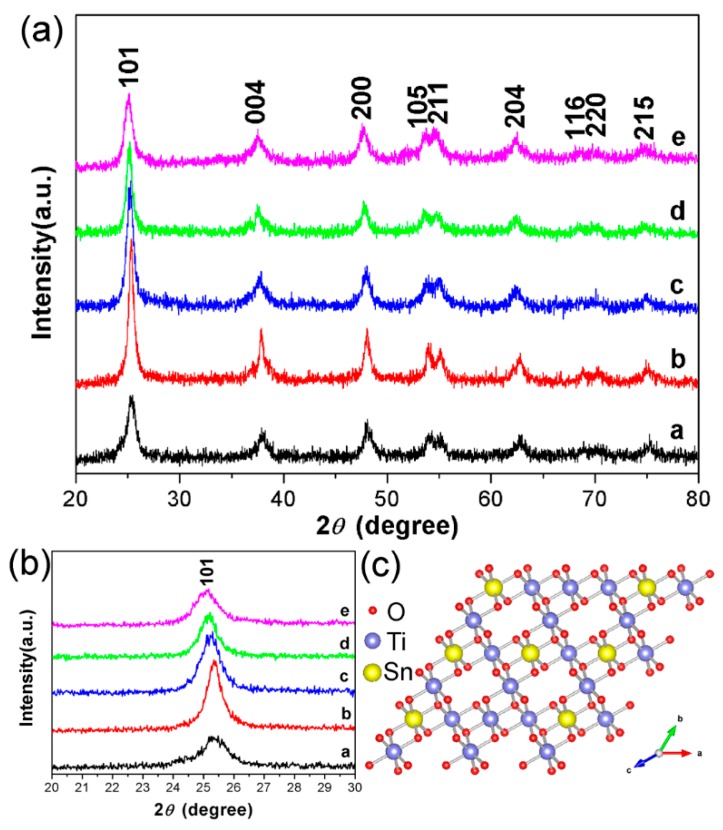
(**a**) The XRD of TiO_2_ hierarchical nanospheres (24 h) (curve a) and STO hierarchical nanospheres with different molar ratios of *n*_Sn_/*n*_Ti_: 1% (curve b), 5% (curve c), 10% (curve d), and 15% (curve e). (**b**) The corresponding XRD magnification image in the range of 2*θ* = 20°–30°. (**c**) Atomic structure model for the STO hierarchical nanospheres.

**Figure 8 nanomaterials-09-01603-f008:**
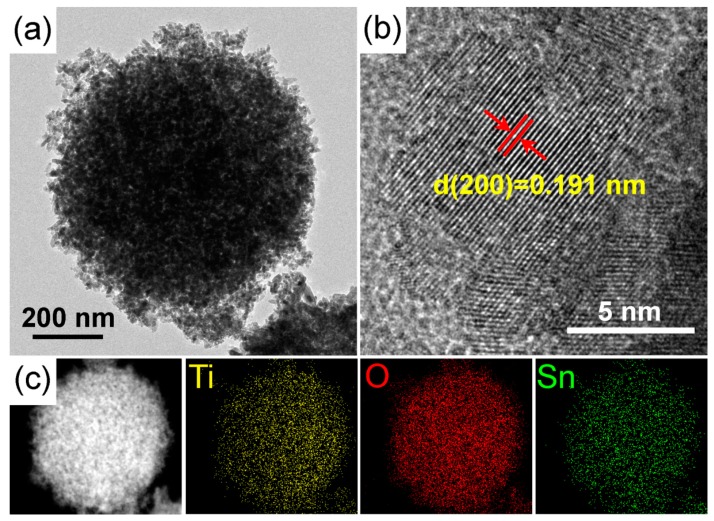
(**a**) TEM and (**b**) HRTEM images of an STO hierarchical nanosphere with a 5% molar ratio of *n*_Sn_/*n*_Ti_. (**c**) The scanning transmission electron micrograph (STEM) images and corresponding STEM-EDX elemental mapping of an STO hierarchical nanosphere.

**Figure 9 nanomaterials-09-01603-f009:**
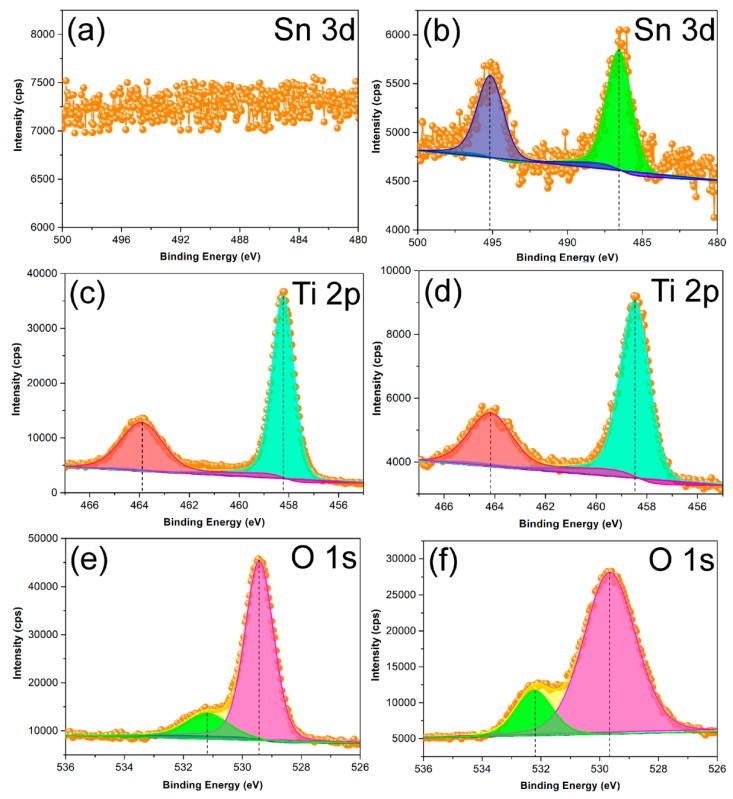
X-ray photoelectron spectroscopy (XPS) results of the TiO_2_ hierarchical nanospheres (24 h) (**a**,**c**,**e**) and STO hierarchical nanospheres with a 5% molar ratio of *n*_Sn_/*n*_Ti_ (**b**,**d**,**f**).

**Figure 10 nanomaterials-09-01603-f010:**
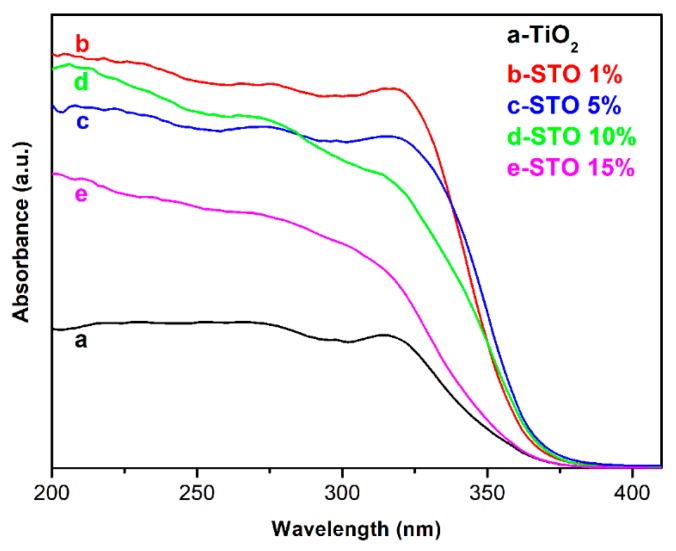
UV-vis absorption spectra of TiO_2_ hierarchical nanospheres (24 h) (curve a) and STO hierarchical nanospheres with different molar ratios of *n*_Sn_/*n*_Ti_: 1% (curve b), 5% (curve c), 10% (curve d), and 15% (curve e).

**Figure 11 nanomaterials-09-01603-f011:**
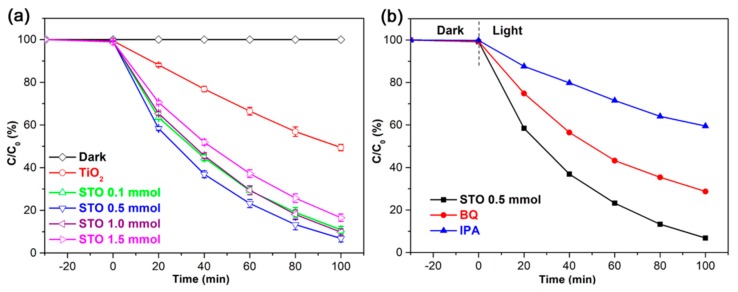
(**a**) Photocatalytic activity of TiO_2_ (24 h) and STO hierarchical nanospheres with different molar ratios of *n*_Sn_/*n*_Ti_: 1%, 5%, 10%, and 15%. (**b**) Photocatalytic degradation of methylene blue (MB) with STO (0.5 mmol) hierarchical nanospheres in the absence and presence of scavengers (BQ and IPA).

**Figure 12 nanomaterials-09-01603-f012:**
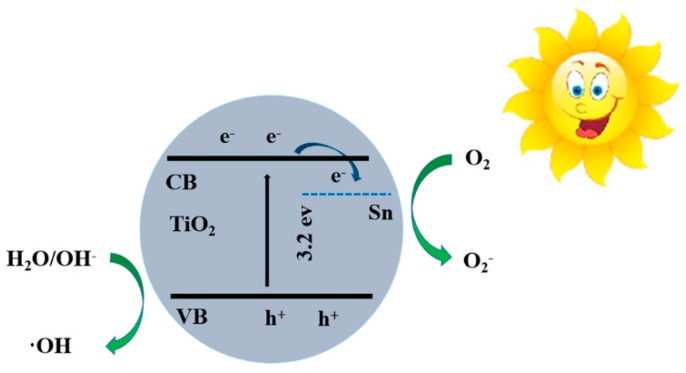
The energy band and photoinduced charge separation and transport in STO hierarchical nanospheres. CB, conduction band; VB, valence band.

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
