# Peer review of "Improving Photocatalytic Degradation Activity of Organic Pollutant by Sn^4+^ Doping of Anatase TiO_2_ Hierarchical Nanospheres with Dominant {001} Facets"

_nanomaterials, 2019, doi:10.3390/nano9111603_

Round 1

Reviewer 1 Report

In this manuscript, the high-energy {001} facets and Sn4+ doping have been demonstrated to be effective strategies to improve the surface characteristics and energy band structures of TiO2 hierarchical nanospheres, thereby improving their photocatalytic performance. The obtained products are thoroughly characterized by XRD, XPS, Raman, SEM, TEM, optical, and catalytic properties. The manuscript is well organized and contains interesting findings. However, I recommended a major revision of the article from its present form before it can be published in nanomaterials. The main concerns are listed below.

The authors should explain the novelty of the present report in a scientific manner? The authors should provide the schematic representation for the experimental procedure. Photoluminescence and trapping experiments are needed to explore the mechanism. The authors should check the loading effect of the catalyst for the degradation of dye. What is the interaction between TiO2 and Sn4+, which make it stable? chemical bond? physical force? how to confirm that heterojunction really form by direct proof.? why is it not just a doping? How to exclude the sensitization mechanism of MB for the present work? TOC is needed to sure the mineralization ratio. The influence of the medium must be studied. Standard deviations of results must be provided (Figures 10).          The latest, important publications are suggested, such as,  J. Environ. Sci. 74 (2018) 107, Ceram. Int. 45 (2019) 5743, etc. In the current state, there are more typographical errors and the language should be improved. Therefore, the authors are advised to recheck the whole manuscript for improving the language and structure carefully.

Reviewer 2 Report

The authors present the fabrication of TiO2 hierarchical nanospheres by solvothermal method applying different reaction times. The resulting TiO2 hierarchical nanospheres expose the high-energy {001} facets and their photocatalytic properties are evaluated, and ascribed to several factors. The TiO2 nanospheres showing the best photocatalytic performance were doped with different Sn4+ doping levels forming SnxTi1-x O2 (STO) nanospheres. Those showed improved photocatalytic performance as compared to non-doped TiO2 nanospheres. The different nanospheres were characterized by several techniques. The manuscript is well organized, the nanostructures were properly characterized, but the results and description of results can be improved, as well as English style. In addition, the literature shows several previous works reporting Sn doped TiO2 nanostructures showing an improvement of photocatalytic performance. That initially limits the novelty of the manuscript, unless the authors justify it solidly. Therefore, I cannot accept the manuscript in the current form to be published in Nanomaterials. Before suggesting the manuscript for publication the authors must address some points.

1-Figure 1: the SEM images must show the same magnification for the sake of a proper comparison. In the Figure 2 the scale bar has not the same length in the different SEM images. Please, correct that.

2-The authors must state in the section 3.2 that the Sn doped nanospheres are based on those ones that showed the best photocatalytic performance, i.e., 24 h of reaction time.

3-Section 3.2: the effect of the different Sn doping level on the TiO2 nanosheets must be added during the description of the corresponding SEM images.

4-Figure 9: The authors described the red-shift of the absorbance for the doped nanospheres. However, they do not explain/justify the remarkable increased of absorption in the UV region for the Sn doped nanospheres as compared to non doped one. What are the reasons of the significant increasing of the UV absorbance? Please, provide a description/explanation of this effect.

5-The authors claim on page 10: “the absorption edges of STO hierarchical nanospheres (curve b-e) shift toward longer wavelengths as compared with the pristine TiO2 UV-vis pattern (curve a).”Please, provide the corresponding values for the sake of clear comparison.

6-The authors claim on page 11:” absorption edges of STO hierarchical nanospheres with a relatively large amount of Sn4+ doping will be blue-shifted…”. If the wavelength shift towards larger wavelengths values, does not it mean the values are red-shifted?

7-Provide the illumination wavelength used during the photocatalytic measurements showed in Figure 4 and 10.

8-On page 12, what do the authors mean by:” After Sn4+ doping, the surface state level of the sample increases…”. Actually, what is the difference between point 3 and 4?

9-Again, on conclusions can you develop more about the meaning of:” greatly improving their energy band structure..”. Is greatly the proper term to define the improvement of energy band structure? By the way, what does improvement of energy band structure mean?

10- The authors must justify/describe the novelty of this work as compare to previous published works is, and stress it into the introduction.

11-A thorough revision of the English style is hardly suggested.

Round 2

Reviewer 1 Report

The manuscript is well organized in the revised version and it can be accepted in the present form. 

Reviewer 2 Report

The comments were addressed and I can recommend the manuscript for publication in Nanomaterials.

However, still English style/typos revision/edition is still certainly needed.